# Study on Tensile Properties of CFRP Plates under Elevated Temperature Exposure

**DOI:** 10.3390/ma12121995

**Published:** 2019-06-21

**Authors:** Yongxin Yang, Yanju Jiang, Hongjun Liang, Xiaosan Yin, Yue Huang

**Affiliations:** 1Central Research Institute of Building and Construction Co. Ltd, Beijing 100000, China; yangyongxin@tsinghua.org.cn; 2The School of Civil Engineering, Wuhan University; Wuhan 430000, China; 2018202100053@whu.edu.cn (Y.J.); hongjunliang8@163.com (H.L.); 13297048175@163.com (Y.H.); 3The School of Civil Engineering and Architecture, Zhongyuan University of Technology, Zhengzhou 450000, China

**Keywords:** CFRP plate, elevated temperature, tensile properties, ANOVA

## Abstract

Elevated temperature exposure has a negative effect on the performance of the matrix resin in Carbon Fiber Reinforced Plastics (CFRP) plates, whereas limited quantitative research focuses on the deteriorations. Therefore, 30 CFRP specimens were designed and tested under elevated temperatures (10, 30, 50, 70, and 90 °C) to explore the degradations in tensile properties. The effect of temperature on the failure mode, stress-strain curve, tensile strength, elastic modulus and elongation of CFRP plates were investigated. The results showed that elevated temperature exposure significantly changed the failure characteristics. When the exposed temperature increased from 10 °C to 90 °C, the failure mode changed from the global factures in the whole CFRP plate to the successive fractures in carbon fibers. Moreover, with temperatures increasing, tensile strength and elongation of CFRP plates decreases gradually while the elastic modulus shows negligible change. Finally, the results of One-Way Analysis of Variance (ANOVA) show that the degradation of the tensile strength of CFRP plates was due to the impact of elevated temperature exposure, rather than the test error.

## 1. Introduction

Fiber Reinforced Plastics (FRP) composites are increasingly applied for strengthening and repairing existed structures in varying forms, including FRP plates, FRP cloths, FRP bars et al. It is divided into carbon fiber reinforced plastics (CFRP), glass fiber reinforced plastics (GFRP), aramid fiber reinforced plastics (AFRP) and basalt fiber reinforced plastics (BFRP) from the nature of the material. This is because FRP materials have advantages of high strength-to-weight ratio, good resistance to electrochemical corrosion, and convenience for installation [1,2,3,4,5]. The FRP material usually needs to be externally bonded to the surface of the concrete structures and steel structures by adhesives. The adhesives are mostly composed of polymeric epoxy resin, which is very sensitive to ultraviolet light, high temperature, humidity and so on. Moreover, epoxy resins are also used to shape FRP fibers when forming FRP plates and FRP bars. Therefore, the degradation of FRP composites in harsh environments is the focus of current research, although FRP fibers can keep their strengths in harsh environments. For the most used carbon fibers, the critical service temperature is between 600 °C and 2000 °C [6,7], but the adhesives change from a brittle-elastic glassy state at low temperatures, to a leathery or rubbery state during heating, and finally a decomposed state (for cross-linked polymers) or viscous liquid state (for linear polymers) at high temperatures [8]. The critical temperature is called the glass transition temperature (*T*_g_). Through using new resins, special curing agents and fillers, the adhesives can resist high temperature (350 °C) and low temperature (−196 °C). However, the glass transition temperature of the epoxy resins used for FRP molding is generally in the range of 55 to 120 °C. The above-mentioned adhesive softening, together with adhesive/fiber bond degradation, leads to a rapid reduction in stiffness and strength of FRP composites. In addition, FRP plates with more layers of fibers are preferred for increasing strength and stiffness; worse degradation in high temperatures is expected because more adhesives are used. Therefore, in practical application for construction, the threat of high temperatures to FRP composites is inevitable [9,10]. Especially for CFRP plates, the mechanical properties after high temperature exposure should be concerned.

Up to now, extensive research has been conducted on mechanical properties of FRP composite or raw materials of FRP composites under elevated temperatures. Bai et al. [11] conducted Dynamic Mechanical Analysis on polyurethane adhesive at different heating rates up to 100 °C and demonstrated that the degradation of polyurethane adhesive modulus depends on the thermal loading history. In other words, adhesive modulus degradation was not only temperature-dependent but also time-dependent. Based on experimental data, temperature- and time-dependent models for physical and mechanical properties at elevated and high temperatures are proposed [12]. Feih et al. [13] studied tensile properties of carbon fibers following exposure to simulated fires of different temperatures (250–700 °C) and atmospheres (air and inert), in which carbon fiber modulus reduced in air and unchanged in inert (above ~500 °C), while tensile strength decreased in both atmospheres (400–700 °C). This indicates fiber modulus was sensitive but fiber tensile strength was insensitive to oxygen content. Cao et al. [14] performed tensile test of CFRP cloths in the range from 20 °C to 120 °C. The *T*_g_ of the structural adhesive used in the test was between 42 °C and 45 °C. Analysis results indicated that tensile strength of CFRP decreased significantly when the test temperature exceeded the glass transition temperature of the structural adhesive. Lu et al. [15] found that elevated temperatures and sustained tensile loading caused irreversible degradation in the fiber–matrix interface of BFRP plates due to stress redistribution, resulting in serious degradation of the tensile properties. The higher the exposure temperature, the more serious was the resulting degradation. Milad et al. [16] observed that the flexural and impact properties of GFRP laminated decrease with the increase of temperature and exposing time. When the temperatures are below *T*_g_ (60 °C), the bending strength of all the laminates were almost resistant, while the impact performance decreased more than 10%. Moreover, the diagnostic plots based on Bayesian models were proposed to predict the tensile strength of CFRP and GFRP sheets, with and without intumescent paint at different temperatures by Majid et al [17]. Hamed et al [18] have conducted research on the performance of FRP bars under fire condition. The tensile strength can remain stable when the exposed temperature was below 90 °C, and decreased significantly under high temperature exposure above 90 °C. Wang et al [19] have presented an experimental study of the mechanical properties of FRP reinforcement bars under high temperatures exposure. It was found that the FRP composite bars retain a very high level (90%) of their original stiffness below 350 °C. Strength reductions of different FRP bars are about 45% and 35% of the original ambient temperature strengths of GFRP and CFRP composite bars respectively at 350 °C.

In conclusion, the above research results established the foundation for studying mechanical properties of FRP composites under elevated temperature exposure, which mainly focuses on FRP sheet, FRP fiber or FRP composite bars. However, the effect of elevated temperature exposure on the mechanical properties of CFRP plates have been little investigated. In this paper, static tensile tests were carried out on CFRP plates under elevated temperature exposure to study the effects of elevated temperature on tensile strength, elastic modulus and the elongation of CFRP plates. 

## 2. Experimental Program

### 2.1. Specimen Preparation

A total of 30 CFRP plate specimens were fabricated and divided into 5 groups as shown in Table 1. This type of CFRP plates chosen was the most commonly used product that has reached industrial level. The mechanical properties of the CFRP plate provided by the manufacturer were listed in Table 2. The high-strength CFRP plates were domestic with a fiber content of 70%. According to GB/T 1447-2005, CFRP plates were cut into 500 mm long × 15 mm wide × 1.2 mm thick as shown in Figure 1. The ends of the FRP samples were attached to two reinforcing aluminous gaskets to apply load. The width, length and thickness of the aluminous gasket were 15 mm, 90 mm, and 2 mm. The aluminum surfaces needed to be roughened to ensure a good fit with FRP before bonding. At the same time, the aluminum gasket must be chamfered to reduce the shear stress and peeling stress near the end of the test piece, thus preventing the specimen from breaking near the end. The adhesives used between aluminum plates and CFRP plates were two-component epoxy resin. Then, subsequent to fully curing for 7 days at room temperature, every CFRP plate specimens was pasted a 6.9 mm × 3.9 mm strain gauge longitudinally in the middle. Considering the high temperatures, heat resisting acrylic adhesive (−55–180 °C) was selected between strain gauges and CFRP plates. 

### 2.2. Test Apparatus

The CFRP plate specimens were conditioned with high temperatures in a heating chamber as shown in Figure 2. The heating chamber was customized according to the test requirements. A hole is provided inside to allow the specimens to pass through. The middle part of the specimens passed through the heating chamber, and the two ends are respectively clamped by the grips of a 30T universal electronic material testing machine (Metis industrial systems co., Ltd, Shenzhen, China) as shown in Figure 3. The accuracy of the load cell is 0.01 kN. The specimens were then heated to the desired temperature (10, 30, 50, 70, and 90 °C) through the heating chamber and held for 20 min. The accuracy of the heating chamber is ±1 °C. Then, the specimens were tested in tensile with a displacement rate of 2 mm/min. The data of the strain gauges were collected by the DH3816N acquisition system (Donghua Testing Technology Co., Ltd, Jiangsu, China). It should be noted that the chamber was pre-heated to determine temperatures and then the specimens were placed in it to ensure the exact temperature and duration. The results of the test are shown in Table 1.

## 3. Results and Discussion

### 3.1. Failure Mode

The CFRP plate specimens exposed to 10 °C and 30 °C conditions showed similar failure modes. As shown in Figure 4a,b, ply delamination was observed and the plate specimens broke into several strips. After pultruded CFRP plates were split, the rest of the plates with continuously bearing loads were suddenly broken. Figure 4c–e exhibit failure modes of CFRP plate specimens subjected to 50 °C, 70 °C, and 90 °C of high temperature exposure. For 50 °C exposure, the CFRP plates in the heated region broke into a mixture of carbon fiber silks and carbon fiber strips. For 70 °C exposure, there were fewer fiber strips, and the failure mode of CFRP plates became completely filamentous when the exposure temperature rose to 90 °C exposure. The failure mode was a burst failure. Obviously, with the applied temperatures arose, the thinner strips or more fiber silks were found when failure. The reason is that the epoxy resin was used to shape FRP fibers when FRP plates were formed. However, the epoxy resin on the CFRP plates softened in varying degrees when the exposure temperature reached or exceeded the glass transition temperature of the epoxy resin. Due to epoxy resin softening which leads to epoxy resin/fiber bond degradation, the carbon fibers could not carry tensile force together. Accordingly, during tensile tests, the carbon fibers fractured one after another. 

### 3.2. Stress-Strain Curves

The stress-strain curves of the specimens were plotted in Figure 5. The shape of the curves under different temperature exposure were linear, which indicates that CFRP plates still maintain good linear elastic characteristic under elevated temperature exposure. The failures of specimens were brittle and sudden without warning so that the curves had no yield plateau. By comparing the curves, it can be seen that the peak stress and ultimate strain of the specimens both changed negligibly as the temperature increased. The influence of elevated temperature on the tensile strength, the elastic modulus and the elongation of the specimens will be analyzed below.

### 3.3. Effect of Elevated Temperature on Tensile Strength of CFRP Plates

Figure 6 plots the variations of tensile strengths of CFRP plate specimens as a function of temperatures. Compared with the case of 10 °C, the tensile strength of 30 °C case showed a negligible increase of 0.22% while that of 50 °C, 70 °C and 90 °C cases reduced by 5.5%, 8.5% and 9.0%, respectively. In the light of previous studies, carbon fibers are immune to high temperatures up to 400 °C. The loss in tensile strength of CFRP plate specimens was attributed to the softening of epoxy resin. The deterioration rate was maximum at 50 °C while relatively stable at 70 °C and 90 °C. The related investigation [14] found the glass transition temperature of the adhesive by the Dynamic Mechanical Analysis (DMA) method between 42 °C and 45 °C. Therefore, it was concluded that the tensile strength of CFRP plates would decrease visibly when the exposure temperature was beyond the glass transition temperature of the adhesive.

### 3.4. Effect of Elevated Temperature on Elastic Modulus of CFRP Plate

Figure 7 shows the elastic modulus of the CFRP plates under different temperature exposure. As can be seen, elevated temperature exposure had little effect on the elastic modulus of CFRP plates. Compared with the specimens under 10 °C, the elastic modulus of specimens under 50 °C, 70 °C, and 90 °C decreased by 0.50%, 0.68% and 0.90%, respectively, which could be ignored. As for the composite CFRP plate, its elastic modulus (*E*_c_) can be calculated as:(1)$$E_{c}=(E_{f}t_{f}+E_{m}t_{m})/t_{c}$$
in which, *E*_f_ and *t*_f_ are elastic modulus and thickness of the fiber, respectively; *E*_m_ and *t*_m_ are elastic modulus and thickness of the resin matrix in specimens, respectively; *t*_c_ is thickness of the specimens. 

The *E*_f_ of the carbon fibers is generally between 160 GPa and 450 GPa, and the *E*_m_ of the matrix resin is about 4.5 GPa [20]. Moreover, the thickness of the resin matrix was considered to be only about 1/11 of the thickness of the whole plate. Therefore, carbon fibers are the dominant factor affecting the elastic modulus of CFRP plates. However, the elevated temperature has little effect on the elastic modulus of carbon fibers [21] so that the decrease of the CFRP plate specimens after elevated temperature is almost negligible. As mentioned earlier, elevated temperatures can cause the adhesive to soften and reduce the synergism between the fibers, resulting in that carbon fibers are unevenly stressed. Thus, the discreteness of the elastic modulus increases.

### 3.5. Effect of Elevated Temperature on Elongation of CFRP Plate

Figure 8 shows the degradation of the elongation of CFRP plates under elevated temperature exposure. With the temperature increasing, the elongation of specimens decreases gradually, and its standard deviation tends to increase. Compared with the specimens under 10 °C, the elongation of specimens at 50 °C, 70 °C, and 90 °C decreased by 0.65%, 5.2%, 7.7% and 9.0%, respectively. The decreasing trend of elongation change is consistent with the tensile strength. The main reason is that the elastic modulus is little affected by the temperature change.

## 4. Discussion of ANOVA

In order to determine whether the change of tensile strength and elastic modulus is caused by the elevated temperature exposure or the random error in the experiments, the one-way One-Way Analysis of Variance (ANOVA) evaluation method was adopted. Assuming factor A is the only variable, it has *n* levels, denoted by *A*_1_, *A*_2_, ⋯, *A*_r_. *n_i_* times independent experiments are made at level *A_i_*, and the obtained sample values were denoted by $\left(x_{i1},x_{i2},\cdots ,x_{i,n_{i}}\right)$. Assume that the level of *A*j are regarded as specimen observations from the *i*th normal distribution $X_{i}\sim N\left(\mu_{i},\sigma^{2}\right)$, where μ*_i_* and σ^2^ are unknown, and each population *X_i_* is independent of each other. The difference in the *r* levels of the comparison factor *A* is attributed to comparing the mean of the *r* populations. 

The British statistician R.A. Fisher discussed this problem from the perspective of variance composition, and gave the following sum of squares decomposition theorem:(2)$$S_{T}=S_{E}+S_{A}$$
in which,
(3)$$S_{E}={\sum_{i=1}^{r}\sum_{j=1}^{n_{i}}\left(x_{\mathrm{ij}}-\bar{x_{i\cdot }}\right)}^{2}$$
(4)$$\bar{x_{i\cdot }}=\frac{1}{n}\sum_{j=1}^{n_{i}}x_{\mathrm{ij}}$$
(5)$$S_{A}={\sum_{i=1}^{r}\sum_{j=1}^{n_{i}}\left(\bar{x_{i•}}-\bar{x}\right)}^{2}=\sum_{i=1}^{r}n_{i}{\left(\bar{x_{i•}}-\bar{x}\right)}^{2}$$
in which, *S_E_* represents the influence of random error, called the sum of squares within the group; *S_A_* represents the sum of the differences between the specimen mean and the total mean at the *A_i_* level, which reflects the difference between the *r* mean values, called the sum of squares between groups. If the null hypothesis *H*_0_ is established, *S*_E_/(*n* − *r*) and *S*_A_/(*r* − 1) are unbiased estimates of σ^2^ by statistical analysis, and *S_E_* and *S_A_* are independent of each other. Therefore, a random variable (F) with degrees of freedom *r* − 1 and *n* − *r* can be constructed as follows:(6)$$F=\frac{S_{A}/\left(r-1\right)}{S_{E}/\left(n-r\right)}\sim F\left(r-1,n-r\right)$$
Therefore, the random variable *F* can be used as the test statistic of *H*_0_. Under the condition of a certain significance level α, if $F\geq F_{\alpha }\left(r-1,n-r\right)$, the null hypothesis is rejected, and the alternative hypothesis is accepted, indicating that the impact of the test factors on the population is significant. If *F* < *F_α_*, the null hypothesis is accepted, indicating that the test factor has no significant effect on the population.

The results of the ANOVA about tensile strength and elastic modulus of the CFRP plates are shown in Table 3. It can be seen that the *F*-value of the tensile strength in the interval of 90% confidence is 2.4, and that of the elastic modulus is 6.29. These values are greater than the critical *F*-value (2.18) under the significance level of *α* = 0.1. So, the null hypothesis *H*_0_ is rejected, indicating that the degradation of tensile strength and the elastic modulus are not caused by the test error. In other words, elevated temperature has a negative effect on the tensile strength and the elastic modulus of CFRP plates with a statistical significance.

## 5. Conclusions

This work describes an experimental investigation of 30 CFRP plates under elevated temperature exposure subjected to axial tension. The following conclusions are reached within the scope of the current research. When the exposed temperature of CFRP plates increased from 10 °C to 90 °C, the failure mode changed from the whole CFRP plate being fractured to the continuous fracture of carbon fibers in CFRP plate. The reason is that the epoxy resin was used to shape FRP fibers when forming FRP plates. However, the epoxy resin on the CFRP plates softened in varying degrees when the exposure temperature reached or exceeded the glass transition temperature of the epoxy resin. Due to epoxy resin softening which led to epoxy resin/fiber bond degradation, the carbon fibers could not carry tensile force together. Accordingly, during tensile tests, the carbon fibers fractured one after another. Meanwhile, with the increase of temperature, the tensile strength and the elongation of CFRP plates decreased obviously, while the elastic modulus remained basically unchanged. Compared with the specimens under 10 °C exposure, the tensile strength and the elongation of CFRP plates under 90 °C decreased about 9.0%, whereas the elastic modulus only decreased by 0.89%. This is because the carbon fibers are the dominant factor affecting the elastic modulus of CFRP plates. After high temperature exposure, the performance of the specimens become more discrete. Finally, ANOVA analysis showed that the elevated temperature exposure had a statistically significant difference on the tensile strength and the elastic modulus of CFRP plates, and it was not caused by the test error. 

## Figures and Tables

**Figure 1 materials-12-01995-f001:**
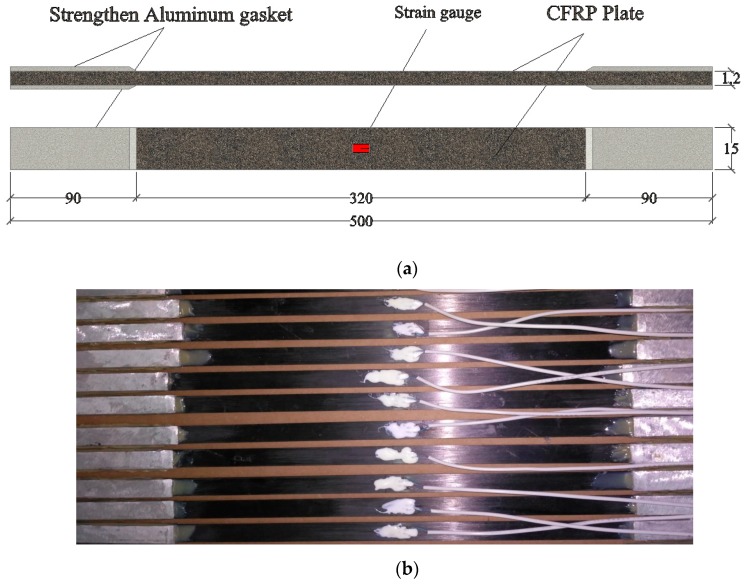
Configuration of Carbon Fiber Reinforced Plastics (CFRP) plate specimens (unit: mm): (**a**) Schematic diagram; (**b**) Photos.

**Figure 2 materials-12-01995-f002:**
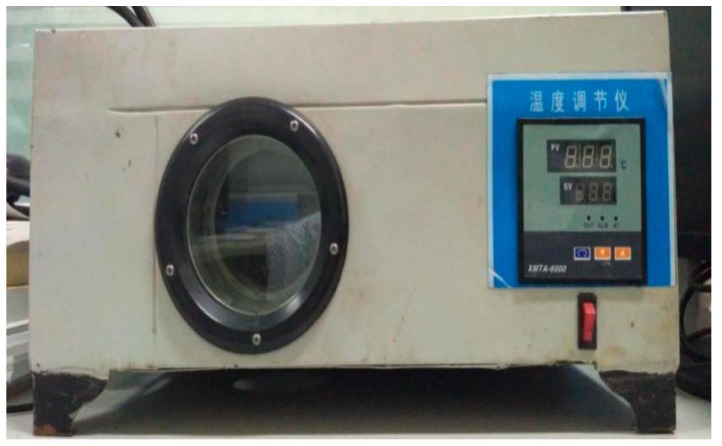
Temperature test chamber.

**Figure 3 materials-12-01995-f003:**
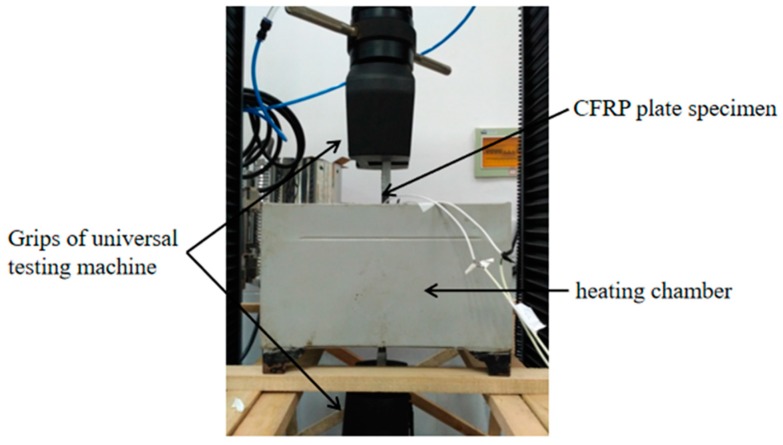
General view of the test setup.

**Figure 4 materials-12-01995-f004:**
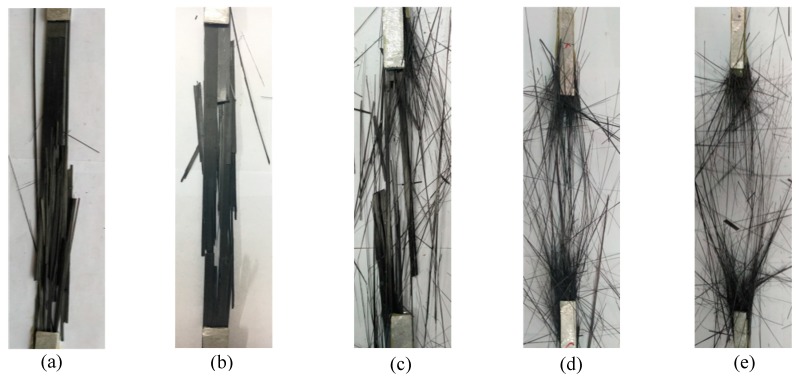
Typical failures of CFRP plates: (**a**) C-T10, (**b**) C-T30, (**c**) C-T50, (**d**) C-T70 and (**e**) C-T90.

**Figure 5 materials-12-01995-f005:**
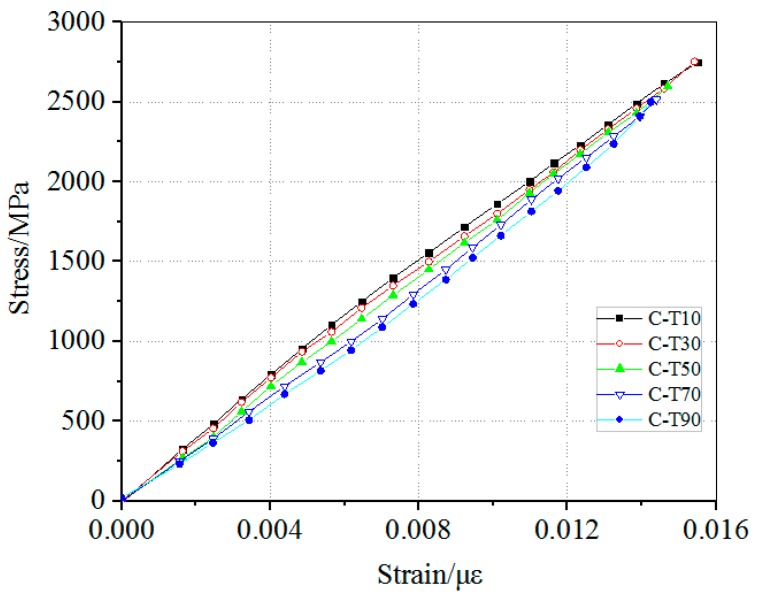
The stress-strain curves of the specimens.

**Figure 6 materials-12-01995-f006:**
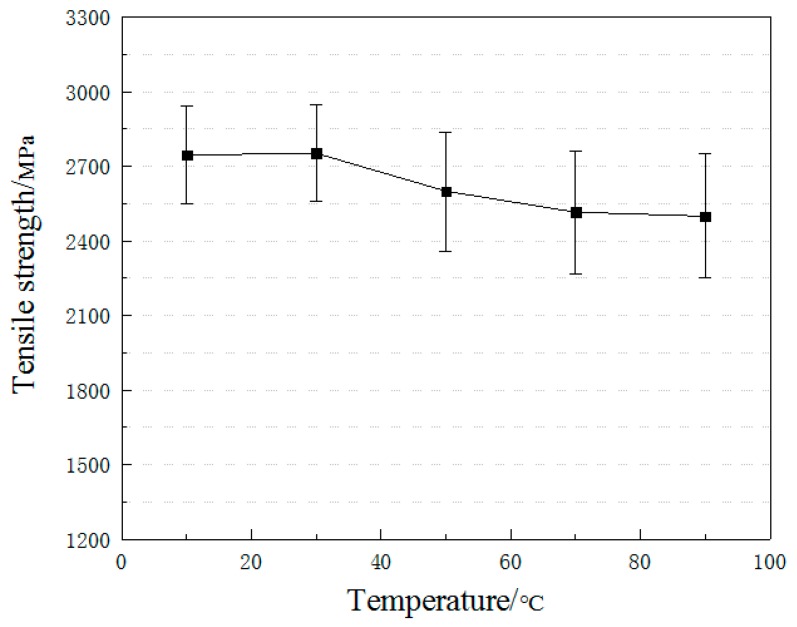
Tensile strength versus temperature.

**Figure 7 materials-12-01995-f007:**
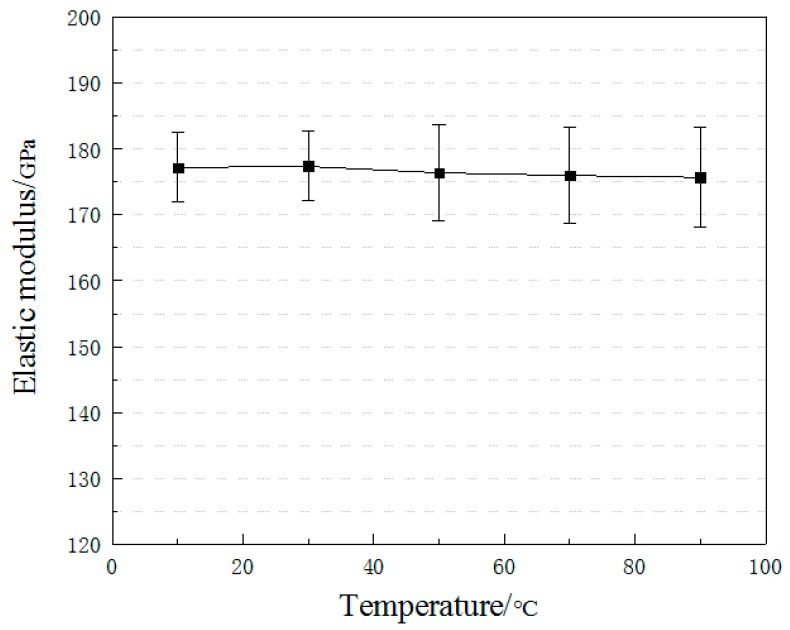
Elastic modulus versus temperature.

**Figure 8 materials-12-01995-f008:**
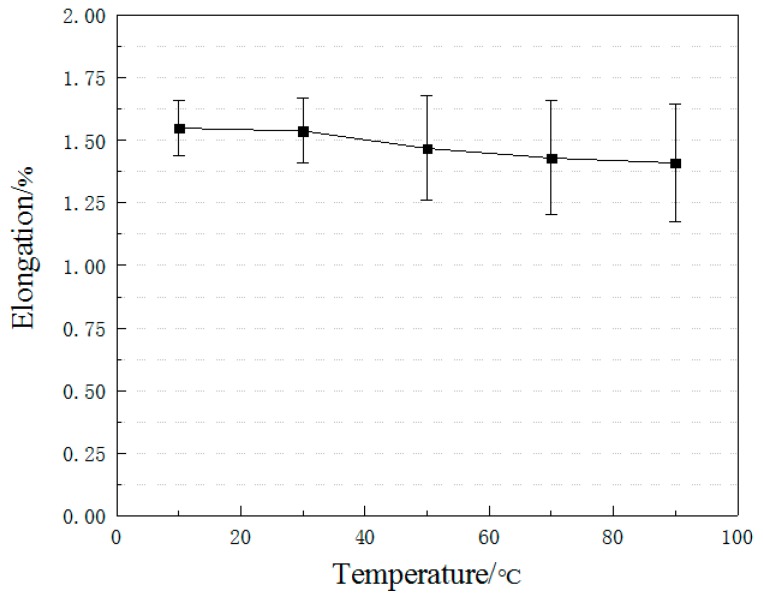
Elongation versus temperature.

**Table 1 materials-12-01995-t001:** The properties of CFRP plate specimens under different temperatures exposure.

Samples	T/°C	Number	Ultimate Load/kN	Ultimate Strength/MPa		Elastic Modulus/GPa		Elongation/%	
				Mean	Standard Deviation	Mean	Standard Deviation	Mean	Standard Deviation
C-T10	10	6	49.78	2748	196.52	177.24	3.71	1.55	0.118
C-T30	30	6	49.57	2754	196.02	177.45	3.84	1.54	0.121
C-T50	50	6	46.76	2597	239.5	176.36	5.04	1.47	0.178
C-T70	70	6	45.28	2515	244.02	176.03	5.35	1.43	0.208
C-T90	90	6	45.02	2501	249.51	175.65	5.57	1.41	0.203

**Table 2 materials-12-01995-t002:** Mechanical properties of CFRP plates provided by the manufacturer.

Performance Indicators	Ultimate Strength (MPa)	Elastic Modulus (GPa)	Elongation (%)
CFRP plates	≥2400	≥160	≥1.4

**Table 3 materials-12-01995-t003:** One-Way Analysis of Variance (ANOVA) results of mechanical property of CFRP plates.

	Sources	Degree of Freedom	Sum of Squares	Mean Squares	*F*-Value	Critical *F*-Value
Tensile strength	Feed	4	360060	90015	2.4	2.18
	Residuals	25	921307	36852	—	—
Elastic modulus	Feed	4	14.415	3.604	6.29	2.18
	Residuals	25	567.125	22.685	—	—

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
