# Peer review of "Study on Tensile Properties of CFRP Plates under Elevated Temperature Exposure"

_materials, 2019, doi:10.3390/ma12121995_

Round 1

Reviewer 1 Report

This is an experimental article based on the tensile test of CFRP specimens subjected to temperature. The article in this case is rejected due to:

- It is recommended to add examples of the use of the material at industrial level and relate it to the problems in the matrix.

- The number of specimens for each temperature condition is appropiate, but the approach of the results based on the ANOVA analysis is insufficient. It is proposed to add information from the ANOVA analysis in the experimental by omitting the development of the equations and going directly to the subsequent relationship with the results. In addition, it is recommended to add the data processing software.

- It is recommended to delete figure 2(b). It does not provide relevant information.

- The results should start with table 1 and the ANOVA analysis should be used as much as possible during the different points exposed to discuss their degree of influence together with the data description performed. Currently, it only describes what happens during the experiments by representing the data without relating their degree of significance.

- In point 3.1 the fracture mode for composites should be mentioned and related to the results.

- Point 3.2 should represent the curve of all tests. Only C-T110 and C-T170 appear.

Author Response

We would like to thank the you for your advice and comments on our manuscript. We have provided a point-by-point response, and the changes have been marked in yellow in the revised manuscript. All of these can be seen in the attachment.

Reviewer 2 Report

The article entitled “Study on Tensile Properties of CFRP Plates under Elevated Temperature Exposure” describes main results of experimental study of CFRP plates’ behaviour under elevated temperatures. The introduction consists of state-of-the-art review in the field of FRP composites’ research. The main part of the article describes the experimental campaign, its setup and discussion of obtained results. Statistical analysis of results’ variation is described and performed in the last part. The authors than conclude the article with main findings of their experimental campaign.

General comments:

The article is quite well structured and written. Its main advantage can be found in the experimental study that provides valuable data of CFRP behaviour (tensile strength, elastic modulus and elongation). On the other hand, there are some minor issues that should be addressed before publishing the article.
First of all, a language correction is required, preferably a proof reading by a native speaker. Although generally the language is quite easy to understand, there are some formulations and sentences that require attention (for examples see Specific comments).
Second, the introduction could be amended with the information about fire resistant epoxy resins (or resins with higher glass transition temperatures). Also, the application to other structures than concrete could be mentioned, as these pose some specific conditions.
Third, the Experimental Program description lacks any information about the materials used in the experiments (what fibres and what resin were used, what are their material properties – tensile strength, modulus of elasticity, elongation, thermal expansion coefficient, glass transition temperature of resin etc.). What were the conditions during test specimens’ preparation (were temperature and moisture in the lab measured) as these are only mentioned as the “room temperature”? The experimental setup could also be described more thoroughly – it is not clear whether the test specimens were temperature conditioned and tested in separate chambers, or if this was just one chamber. Also, how and where were the temperatures measured. It seems than no temperature was measured on the actual specimen (not during the conditioning, nor during the tensile testing).
Fourth, considering the “room temperature” around 20-22 °C, the first test set (C-T10) was, in fact, tested not under elevated, but rather under reduced temperature. Is should be commented in the paper together with the information, how or if were the properties (esp. length of the specimen) monitored during the conditioning or if this was not relevant (one would assume some shrinkage/elongation of the specimen due to the temperature conditioning before testing – but maybe the authors can explain why it has no impact on the results obtained).
Fifth, the discussion of the results should be extended. All relevant results should be published and discussed – for example stress-strain curves of other test sets than C-T10 and C-T70 are missing (also, are these the mean curves, or curves from just one of the specimen from test set?). Broader discussion of the obtained values should follow (for example, why does the elongation decreases with the elevation of temperature etc.).
Sixth, the statistical evaluation of variation of obtained results could be made more interesting for the reader by some general introduction about other possible methods, that could have been used and why did the authors chose ANOVA. Also, were all the results analysed in this way, or was just the tensile strength considered – confidence values of elastic modulus and elongation results should also be mentioned and discussed. Consider adding numbering for each equitation in paper.
Finally, the Conclusions should state the main findings of the research together with some explanation of the results.

Overall, after addressing some issues mentioned bellow, the article could be a valuable and relevant contribution in the field of FRP behaviour research.

Specific comments:

Language and typos corrections (examples) - lines 42-43 “In addition, FRP plates with more layers of fibers are preferred for increasing strength and stiffness, worse degradation in high temperatures is expected because more adhesives are used” – verb missing, “worse degradation”?
lines 66-67 “When the temperatures below Tg (60 °C), the bending strength of all the laminates were almost resistant, while the impact performance decreased more than 10%” – verb missing, “strength…almost resistant”?
lines 117-118 “For 70 °C exposure, the fiber strips were less, and the failure mode…” – “strips were less”?
line 226 “…the elongation of CFRP plates decreased obviously…” – “obviously”?

Line 193 – reference should be added (R.A. Fisher [X])

Figure numbers  from page 4 should be corrected (there are two Fig. 1s)

Author Response

(The authors gave the same response as above.)

Reviewer 3 Report

The paper describes some experimental studies on the influence of elevated temperature exposure on tensile properties of CFRP plates. This type of composite is widely used as a strengthening material in civil engineering because of its high strength-to-weight ratio, good resistance to electrochemical corrosion, ease of installation. One of the major concerns with using FRP reinforcing plates in building construction is their early loss of strength and stiffness at elevated temperatures. Therefore the subject is undoubtedly of interest to the scientific and engineering community.

The described studies are interesting. The results can be used in design and numerical modeling of RC structure members strengthened with CFRP plates (eg externally bonded strengthening systems or near surface reinforcement). Unfortunately, the presentation should be improved. In this form the manuscript should not be published. The following aspects should be clarified/checked:

1.       lines 27-75 - Introduction – the state of the art presented in the introduction is quite limited. There are other publications, not mentioned in this paper, which consider the problem of the influence of elevated temperature on the mechanical properties of FRP composites eg:

Hamed Ashrafi et al, The effect of mechanical and thermal properties of FRP bars on their tensile performance under elevated temperatures, Construction and Building Materials 157 (2017) 1001–1010,

 Y.C.Wanga, P.M.H.Wonga, V.Kodurb,  An experimental study of the mechanical properties of fibre reinforced polymer (FRP) and steel reinforcing bars at elevated temperatures, Composite Structures, Volume 80, Issue 1, September 2007, Pages 131-140

2.       lines 74-76 – Motivation for the research – Authors wrote that “…In this paper, static tensile tests were carried out on CFRP plates under elevated temperature exposure to study the effects of elevated temperatures on tensile strength, elastic modulus and the elongation of CFRP plates...” It is general statement but only one type of CFRP plate was used in the research. Why didn’t you use other resins with higher glass transition temperature for comparative purposes? Please explain why this specific type of CFRP plate was chosen?

3.       lines 77-110 – Specimen preparation – Please give some more information about resin and fibers – at least their mechanical properties in ambient and elevated temperature.

4.       line 87 – Are the dimensions of strain gauge correct?

5.       line 96 – Table 1 – please explain why this range of temperatures was chosen?, Why some of the samples weren’t tested in laboratory (room) temperature (20oC)? What code/method was used to determine the Young’s Modulus?  The standard deviation and coefficient of variation for all tested parameters should be given in the table. Did you give in this table mean values?

6.       line 102 – “…displacement rate…” instead of “…loading rate…”

7.       line 98 – test apparatus – please give some information about the resolution and accuracy of apparatus used during the tests?

8.       lines 107 – should be Figure 2,

9.       line 110 – should be Figure 3; (b) is written twice; side elevation is not necessary; general view of the test setup will be interesting,

10.   line 126 – should be Figure 4; please check also the caption – (d) is written twice; Please check the captions under the photos of failure modes if the order is correct– because in literature samples tested in ambient temperature failed in “burst failure” mode similar to the modes given in figure 4 d and e (Pangang Wu et al, Influences of long-term immersion of water and alkaline solution on the fatigue performances of unidirectional pultruded CFRP plate, Construction and Building Materials 205 (2019) 344–356).

11.   line 137 – should be Figure 5 –stress-strain curves for all tested specimens should be presented,

12.   line 150 – should be Figure 6

13.   line 169 and 179 – should be Figure 7 and Figure 8

14.   line 172 – “…that…” should be deleted – now this sentence is not clear

15.   line 218 – should be Table 2

16.   line 220 – the conclusions are too general. Only one type of CFRP composite was tested so it should be clearly stated that the conclusions are valid only for this type of composite and for generalization, further research have to be done.

Author Response

(The authors gave the same response as above.)

Round 2

Reviewer 1 Report

Accepted after modifying the changes proposed in the first revision

Author Response

-Reviewer 1

Accepted after modifying the changes proposed in the first revision

Response: The authors deeply appreciate the reviewer’s approval.

Reviewer 3 Report

Authors improved the previously reviewed paper. Most of reviewer’s comments given in the first review have been taken into account, but unfortunately, there still are some aspects, which should be corrected/explained:

1.       Section 1 (Introduction) or section 2.1 (specimen preparation) - Please explain in final version of your paper why this specific type of CFRP plate was chosen?

2.       Lines 89-90 – Were the CFRP plates homemade or produced by manufacturer? It is not clear, because in one place Authors wrote that “…CFRP plates were homemade…” and one line earlier it was written that “the performance indicators of the CFRP plates provided by manufacturer…” ?

3.       Line 89 and Table 2 –  please use “…mechanical properties…” instead of “… performance indicators…”

4.       Section 2.2 – please provide load cell accuracy.

5.       Line 113 and Figure 3 – please use “grips” instead of “chuck”

6.       Lines 135 and 240 – the phrase “…quitted working…” is not clear.

7.       Section 3.2 and Figure 5 – please check this section because together with figure 5 it is not clear. It seems that Authors described other figure than Figure 5.

8.       Figure 5 – curves for all tests should be presented. Please, check the titles of the axes.

9.       Equation (1) (between lines 172 and 173) – please, check and correct indexes in the equation (1) and in the text below. In the equation “r” in Er should be a subscript. In this equation you use Er and tr but in the text below Em and tm is used.

10.   Line 196 – “tensile strength and elastic modulus” instead of “tensile strength” – Authors analyzed these two parameters so they should be mentioned at the beginning of this section.

Author Response

A point-by-point response to the reviewer's comments has been attached.
